

# Boundary chaos: Spectral form factor

Felix Fritzsch[1,2★] and Tomaž Prosen[1]

**1** Physics Department, Faculty of Mathematics and Physics,
University of Ljubljana, Ljubljana, Slovenia
**2** Max Planck Institute for the Physics of Complex Systems, Dresden, Germany

★ fritzsch@pks.mpg.de

## Abstract

Random matrix spectral correlations is a defining feature of quantum chaos. Here, we study such correlations in a minimal model of chaotic many-body quantum dynamics where interactions are confined to the system's boundary, dubbed *boundary chaos,* in terms of the spectral form factor and its fluctuations. We exactly calculate the latter in the limit of large local Hilbert space dimension $q$ for different classes of random boundary interactions and find it to coincide with random matrix theory, possibly after a non-zero Thouless time. The latter effect is due to a drastic enhancement of the spectral form factor, when integer time and system size fulfill a resonance condition. We compare our semiclassical (large $q$) results with numerics at small local Hilbert space dimension ($q = 2, 3$) and observe qualitatively similar features as in the semiclassical regime.



# 1  Introduction

The notion of chaos in quantum systems is intimately tied to random matrix theory [1–5], most prominently on the level of spectral statistics [6–8]. A hallmark feature of chaotic systems and the corresponding random matrix ensembles is the presence of correlations in their spectra. Those are conveniently described by the spectral form factor (SFF) [4], which probes correlations in the spectrum on all energy scales and which has recently received a lot of attention in various fields, including, e.g., high-energy physics [9–12] and condensed-matter theory [13–33]. Historically, the SFF has first been shown to agree with the random matrix result for quantum systems with an underlying chaotic classical limit by means of semiclassical methods [34–37]. Henceforth, random matrix spectral correlations have become a widely accepted definition of quantum chaos also in the absence of a classical limit, including for instance lattice systems with only a few states per local site, e.g., spin-chains. To similarly link spectral correlations of such systems with random matrix theory, novel tools had to be developed. They led to the introduction of several exactly solvable models for many-body quantum chaos. Most notably, this includes quantum circuit models which allow for explicitly obtaining the SFF either in the limit of large local Hilbert space dimension [17–21] or in the thermodynamic limit [13–16]. Exact results in such models are not limited to spectral statistics, but also include the computation of correlation functions [17, 38–41], aspects of (deep) thermalization [42–45] as well as the growth of entanglement of states [17, 46–53] and local operators [54, 55] and hence provide valuable insight into the dynamics of generic many-body quantum systems.

    A particularly simple solvable model, recently introduced by the present authors and dubbed *boundary chaos*, is built from a non-interacting, free quantum circuit, in which chaos and ergodicity is induced by locally perturbing the system with an interacting gate (impurity interaction) at its boundary [56]. This setting can be interpreted as a minimal model for integrability breaking due to local perturbations. Such scenarios have been intensively studied in case of interacting integrable systems in the Hamiltonian setting, in which perturbations lead to, e.g., a crossover of spectral statistics [57–65] or the onset of thermalization [59,60,63,64,66–68]. In contrast to those results, the underlying unperturbed model here is free, i.e., non-interacting, and even non-dispersive, and yet for suitable choices of the impurity interaction, this setting exhibits random-matrix spectral statistics, ergodic dynamics [56] as well as entanglement growth at maximum speed for both initial product states and local operators [69]. This indicates, that it is the local perturbation, which restores ergodicity and induces chaos, whereas interactions in the unperturbed model are not a necessary condition for this to occur. As the free dynamics of the unperturbed model survives in the bulk, it can be integrated out analytically. This allows for exact results as it reduces the dimensionality and complexity of the problem in analogy to the concept of a Poincaré surface of section in classical dynamics [70] and the associated quantum transfer operator [71,72].

In the present paper, we demonstrate, that the system is indeed chaotic in the sense of spectral statistics. More precisely, we study the SFF and show its agreement with the corresponding random matrix result. As indicated above, we reduce the many-body problem to an effective two-body problem, which allows for obtaining the SFF and all its moments exactly in the so-called semiclassical limit of large local Hilbert space dimension. For different classes of impurity interactions, i.e., either generic (Haar random) two-qudit gates or T-dual gates [73], which remain unitary under partial transpose, we find the SFF and all its moments to agree with random matrix theory in this limit. In case of generic impurity interactions this holds on all time scales, whereas for T-dual impurity interactions the random matrix result is approached only after some non-universal initial time scale, the so called (many-body) Thouless time. Non-zero Thouless time turns out to be a consequence of a strong enhancement of the SFF if time and system size obey a resonance condition. The latter refers to them sharing a large common factor. We contrast the semiclassical results with numerical results at small local Hilbert space dimension. Surprisingly, many of the observed features in the latter case coincide with the semiclassical results. In particular, the SFF as well as its moments agree with the random matrix result after some initial time, demonstrating that the system is indeed quantum chaotic also far away from the above semiclassical limit. Moreover, the enhancement of the SFF when time and system size are in resonance, i.e, they have a large common factor, persists also for small local Hilbert space dimension for T-dual impurity interactions. In case of generic impurity interaction we observe such enhancement as well, when time is an integer multiple of system size, which is in contrast with the semiclassical results. That is, at small local Hilbert space dimension, we observe a non-zero Thouless time for both classes of impurity interactions, which scales at most linearly with the system size.

In the following we first briefly introduce the SFF in Sec. 2 as well as the boundary chaos circuit in Sec 3. Subsequently, we present the exact semiclassical results in Sec. 4 and discuss numerical results in Sec. 5 before concluding in Sec. 6

## 2 Spectral form factor

The SFF $K(t)$ is a well established measure of spectral correlations in complex quantum systems, which is, in contrast to other spectral statistics, e.g., the level spacing distribution, well suited for an analytical treatment. In the past, this allowed for establishing the quantum chaos conjecture by explicitly computing the SFF from periodic orbits in an underlying chaotic classical system [34–37]. While in recent years the focus shifted to the many-body setting, often without an underlying classical limit, the significance of the SFF persists, as it is amenable to analytic calculations in certain solvable models of many-body quantum chaos [13–21]. There, it illustrates the emergence of universal random matrix behavior in structured many-body systems, e.g. spatially extended systems subject to local interactions. Moreover, the SFF probes spectral correlations on all (quasi-)energy scales and is therefore a more sensitive indicator of quantum chaos.

The SFF is obtained from the connected two-point correlation function of the spectral density via Fourier transform with respect to the (quasi-)energy difference between two levels. This renders the SFF a time-dependent quantity, whose value at time $t$ indicates the strength of correlations between levels with (quasi-)energy difference $\sim 2\pi/t$. While for autonomous systems the time variable $t$ takes continuous values, Floquet dynamics require discrete, integer times. In this work we focuss on the latter only. There, the SFF has a convenient representation in terms of the time evolution operator $\mathcal{U}$ of the system under consideration given by

$$K(t) = \left\langle \left| \mathrm{tr}\left(\mathcal{U}^t\right) \right|^2 \right\rangle - N^2 \delta_{t0} = \left\langle \mathrm{tr}\left(\mathcal{U}^t\right) \mathrm{tr}\left(\mathcal{U}^{-t}\right) \right\rangle - N^2 \delta_{t0}. \tag{1}$$

Here, the brackets indicate an average of an ensemble of similar systems, which is necessary due to the lack of self averaging of the SFF [74]. For convenience we take Eq. (1) as the definition of the SFF. As it involves the average over an ensemble, one might study the distribution of the SFF in terms of its moments $K_m(t)$ given by

$$K_m(t) = \left\langle \left| \mathrm{tr}\left(\mathcal{U}^t\right) \right|^{2m} \right\rangle - N^{2m} \delta_{t0} \,. \tag{2}$$

For Floquet systems which lack time-reversal invariance or any other anti-unitary symmetry, the appropriate random matrix ensemble to compare with is the circular unitary ensemble CUE($N$) given by the unitary group U($N$) equipped with the normalized Haar measure. For the CUE($N$) the SFF reads [4]

$$K(t) = \min\{t, N\} \,, \tag{3}$$

and is exponentially distributed with moments [75]

$$K_m(t) = m! K(t)^m \,. \tag{4}$$

The plateau $K(t) = N$ for times larger than the Heisenberg time $t = N$ signals the fact, that the spectrum is finite. It is present for uncorrelated Poissonian spectra as well. In contrast the initial linear ramp $K(t) = t$ indicates correlations in the spectrum. Hence the presence of this linear ramp in a physical system indicates random-matrix like spectral correlations and consequently justifies calling the system (quantum) chaotic. For most chaotic physical systems the linear ramp is approached only after some initial non-universal dynamics depending on the details of the system. The time after which the SFF coincides with the linear ramp and which marks the onset of universal random-matrix dynamics is called the (many-body) Thouless time $t_{\mathrm{Th}}$.

## 3 Boundary chaos circuit

In this section we introduce the model studied in this work. We moreover explain how the computation of the SFF in this interacting many-body system can be reduced to an effective two-body problem. We consider a Floquet system with unitary evolution operator $\mathcal{U}$ given by a brickwork quantum circuit $\mathcal{U} = \mathcal{U}_2 \mathcal{U}_1$ built from two layers $\mathcal{U}_1$ and $\mathcal{U}_2$. The circuit acts on a chain of qudits, i.e., $q$-level systems of length $L + 1$. The $q$-dimensional local Hilbert space at lattice site $x$ is $\mathcal{H}_x \cong \mathbb{C}^q$ while the total Hilbert space $\mathcal{H} = \otimes_{x=0}^{L} \mathcal{H}_x \cong \mathbb{C}^N$ is of dimension $N = q^{L+1}$. The individual layers are mostly composed from swap gates $P$ which are non-interacting and give rise to free dynamics, whereas a single impurity interaction $U$, i.e., an interacting two-qudit gate at the boundary renders the system quantum chaotic. More precisely the layers of the circuit read

$$\mathcal{U}_1 = \prod_{i=1}^{\lfloor L/2 \rfloor} P_{2i-1,2i} \,, \qquad \text{and} \qquad \mathcal{U}_2 = U_{0,1} \prod_{i=1}^{\lfloor (L-1)/2 \rfloor} P_{2i,2i+1} \,. \tag{5}$$

Here $G_{i,j}$ denotes the 2-qudit gate $G = U, P$ acting at sites $i, j$. A diagrammatic representation of the circuit is given by

$$\mathcal{U} = \qquad \tag{6}$$

where the wires carry the $q$-dimensional Hilbert space $\mathbb{C}^q$ and the local gates are

$$P = \times \,, \quad \text{and} \quad U = \blacksquare \,. \tag{7}$$

The above diagrammatic representation of $\mathcal{U}$ allows for reducing the many-body problem of computing $\text{tr}(\mathcal{U}^t)$, occurring in Eq. (1), to an effective two-body problem. This is best illustrated by the diagrammatic expression

$$\text{tr}(\mathcal{U}^t) = \qquad\qquad\qquad\qquad\qquad\qquad\qquad\qquad (8)$$

depicted here for $L = 5$ and $t = 4$. There, the upper left leg of the impurity interaction at time step $r$ is connected to the lower left leg of the impurity interaction at time step $r + 1$. In contrast, the upper right leg of the impurity interaction at time step $r$ is connected to the lower left leg of the impurity interaction at time step $(r + L) \mod t$. The latter is a consequence of the wires in the non-interacting bulk traversing the system twice, once in forward and once in backward spatial direction, during a time interval of length $L$, and the periodicity in $t$ is due to taking the trace. This fully captures the free bulk dynamics and one might think of this as integrating out the latter and to reduce the dimensionality of the problem from $(1 + 1)$ to $(0 + 1)$ dimensions, which bears some analogy with the concept of Poincaré surface of section in classical dynamics, e.g., in billiard systems. To express this more formally, we denote the canonical product basis in $\mathcal{H}$ by $|i_0 i_1 \cdots i_L\rangle$ and the corresponding basis in the bipartite system $\mathcal{H}_0 \otimes \mathcal{H}_1$ by $|ij\rangle$ with $i, j \in \{0, 1, \ldots, q-1\}$. The diagrammatic argument above then translates into

$$\text{tr}(\mathcal{U}^t) = \sum_{i_0,\ldots,i_L} \langle i_0 \cdots i_L | \mathcal{U}^t | i_0 \cdots i_L \rangle = \sum_{i_1,j_1,\ldots,i_t,j_t} \prod_{s=0}^{t-1} \langle i_s j_s | U | i_{s+1} j_{s+L} \rangle . \qquad (9)$$

The second equality follows from inserting $t - 1$ resolutions of identity between each of the $t$ factors $\mathcal{U}$ and using the definition of the swap gates. A compact notation of the right hand side of the above equation can be obtained by the replica trick, i.e., by interpreting the product as a matrix element of the unitary operator $U^{\otimes t}$ acting on a lattice in time, i.e., the Hilbert space $(\mathcal{H}_0 \otimes \mathcal{H}_1)^{\otimes t} \cong \mathcal{H}_0^{\otimes t} \otimes \mathcal{H}_1^{\otimes t}$. We denote the canonical product basis of $\mathcal{H}_l^{\otimes t}$ by $|\mathbf{i}\rangle = |i_1 \cdots i_t\rangle$ and $|\mathbf{j}\rangle = |j_1 \cdots j_t\rangle$ for $l = 0$ and $l = 1$, respectively, and write $|\mathbf{ij}\rangle$ for the product basis in $\mathcal{H}_0^{\otimes t} \otimes \mathcal{H}_1^{\otimes t}$. To further simplify notation, we consider the representation of the symmetric group on $t$ elements $S_t$, which permutes tensor factors in $\mathcal{H}_l^{\otimes t}$ ($l = 1, 0$). We denote the action of $\sigma \in S_t$ on a basis vector $|\mathbf{i}\rangle$ by $|\sigma(\mathbf{i})\rangle$ Moreover, the periodic shift (mod $t$) by $x$ is denoted by $\eta_x \in S_t$. The above definitions allow for rewriting Eq. (9) and an analogous equation for $\text{tr}(\mathcal{U}^{-t})$ as

$$\text{tr}(\mathcal{U}^t) = \sum_{\mathbf{i},\mathbf{j}} \langle \mathbf{ij} | U^{\otimes t} | \eta_1(\mathbf{i}) \eta_L(\mathbf{j}) \rangle , \quad \text{and} \quad \text{tr}(\mathcal{U}^{-t}) = \sum_{\mathbf{k},\mathbf{l}} \langle \mathbf{kl} | (U^\star)^{\otimes t} | \eta_1(\mathbf{k}) \eta_L(\mathbf{l}) \rangle , \qquad (10)$$

with $U^\star$ being the complex conjugate matrix of $U$ after possibly reordering tensor factors in $\mathcal{H}_i^{\otimes t}$. Consequently, we obtain

$$\left| \text{tr}(\mathcal{U}^t) \right|^2 = \sum_{\mathbf{i},\mathbf{j},\mathbf{k},\mathbf{l}} \langle \mathbf{ij} | U^{\otimes t} | \eta_1(\mathbf{i}) \, \eta_L(\mathbf{j}) \rangle \langle \mathbf{kl} | (U^\star)^{\otimes t} | \eta_1(\mathbf{k}) \, \eta_L(\mathbf{l}) \rangle , \qquad (11)$$

whose ensemble average (over ensembles of impurity interactions $U$ defined below) yields the SFF. The above expression is well suited for further analytical treatment as it essentially constitutes a two-body problem. In contrast, evaluating the expression numerically requires

exponential memory in $t$ and is not suitable for numerical simulations as all but the shortest time scales are not accessible. Furthermore, an interesting observation from the above expression is the periodicity in $L$ for fixed time $t$ following from $\eta_L = \eta_{L+t}$. This implies that the SFF at time $t$ for arbitrary system size $L$ is given by the SFF at time $t$ for system size $L \bmod t$. However, this does not allow to simplify the computation of the SFF on the time scales $t > L$ one is typically interested in.

## 4 Exact SFF in the semiclassical limit

The reduction to an effective two-body problem resulting in Eq. (11) allows for computing the SFF in the semiclassical limit of large local Hilbert space dimension $q \to \infty$ for suitable ensembles of impurity interactions. A natural choice is to take the latter to be a Haar random unitary from $U(q^2)$. Another choice of impurity interactions is to choose them T-dual [73], meaning their partial transpose remains unitary. Both classes lead to ergodic dynamcis [56] and we consider them both in this section. We only state and discuss our results in the following and refer to App. A for an explicit derivation.

### 4.1 A toy model

We start, however, with the trivial example of a non-interacting impurity given by $U = u \otimes v$ as this provides some intuition for the interacting case. Here $u$ and $v$ are two independent Haar-random single qudit gates drawn from $U(q)$. In this situation the trace

$$\mathrm{tr}\left(\mathcal{U}^t\right) = \mathrm{tr}\left(u^t\right)\left[\mathrm{tr}\left(v^{t/n}\right)\right]^n, \tag{12}$$

factorizes with $n = \gcd(t, L)$. This can be seen from Eq. (10) and by noting that the sum over the basis staes $\mathbf{i}$ on the site 0 produces $\mathrm{tr}\left(u^t\right)$. The second factor follows from the permutation $\eta_L$ being a product of $n$ disjoint cycles of length $t/n$. Each of the $n$ cycles produces a factor of $\mathrm{tr}\left(v^{t/n}\right)$ in the sum over the basis states $\mathbf{j}$. An analogous factorization occurs for $\mathrm{tr}\left(\mathcal{U}^{-t}\right)$ as well. Computing the Haar average over the two unitaries $u$ and $v$ in Eq. (11) subsequently determines the SFF for this non-interacting toy model. The boundary's contribution (the average over $u$) to the total SFF is the random matrix SFF $K(t) = t$ of a single particle system with Hilbert space dimension $q$, whereas the bulk of the system (the average over $v$) contributes with higher moments of the SFF $K_n(t/n) = n!(t/n)^n$ of such a single particle system, but at a reduced time $t/n$. The above considerations thus yield the SFF as

$$K(t) = \frac{n!}{n^n} t^{n+1}, \tag{13}$$

for times $t \le q$. Noting that $n!(t/n)^n > t$ if $t > 2n$, the SFF of the toy model will be enhanced compared to both the random matrix result for a single large CUE ensemble, $K(t) = t$, and to that of two independent CUE, $K(t) = t^2$. Furthermore, for fixed $t > 2$ the term $n!(t/n)^n$ is monotonically increasing in $n$, implying that the enhancement of the SFF is more pronounced, if $t$ and $L$ share a large common factor. Hence, the SFF will be largest for times $t$, which are integer multiples of system size, i.e. $n = L$. The above statement could readily be generalized to higher moments $K_m(t)$, yielding the same phenomenology. Moreover, these arguments indicate an intricate interplay of spectral correlations and the common factors of $t$ and $L$. Stressing once again the analogy to classical Poincaré sections this might be interpreted as a resonance condition between the free bulk dynamics and the dynamics along the boundary. The fate of these observations, when replacing the non-interacting impurity with an interacting one, remains to be investigated in the following. The enhancement described above will turn out to be present in most cases accessible by either analytical or numerical methods.

## 4.2 Haar random impurity interactions

We start with the only exception to the previous statement, which is the SFF for Haar random impurity interactions in the semiclassical limit. That is, the average in Eq. (1) is taken over $U(q^2)$ with respect to the Haar measure. By linearity of the expectation value we might take the average of each term in the $4t$-fold ($4mt$-fold in case of the $m$-th moment) sum in Eq. (11) individually. For such terms, i.e., monomials of degree $2t$ ($2tm$) in the matrix elements of $U$ and $U^\star$ there exists a general theory of integration with respect to the Haar measure based on the representation theory of both the symmetric group $S_t$ ($S_{tm}$) and $U(N)$ which yields the Haar average in terms of Weingarten functions Wg [76, 77]. The later are functions defined on the symmetric group and for fixed $\sigma \in S_t$ its value $\text{Wg}(\sigma)$ is a rational function of $N$ with known asymptotics for large $N$ [76–78]. Here, $N = q^2$ and the large $q$ asymptotics has been used to compute the SFF in the random phase circuit of Ref. [17, 18] and the random matrix ensembles of Ref. [21]. Adapting those methods to the case at hand is straitghtforward and carried out in more detail in App. A.1. Here we only state the results. In the semiclassical limit $q \to \infty$ at fixed time $t$ we find both the SFF $K(t) = K_1(t)$ and all its moments $K_m(t)$ to follow the random matrix result for the CUE. That is, we have

$$K_m(t) = m! t^m \,, \tag{14}$$

for all times $t$ and all $m$ and independent of $n = \gcd(L, t)$. In particular the free bulk dynamics of the system has no effect on the spectral correlations in this limit. Note that the plateau of the SFF for times larger than the Heisenberg time cannot be reproduced in this limit as Heisenberg time diverges as $q^{L+1}$ in the semiclassical limit. In fact, at finite $q$ those results should be expected to give the leading contribution to the spectral form factor only up to time scales $t < q$ [21].

The random matrix SFF for all time scales at large $q$, i.e., the system being fully chaotic in the sense of spectral statistics, is in contrast with persistent revivals of correlation functions between local observables [56] implying slow thermalization as well as with slow growth of (local operator) entanglement [69]. Note, however, that those results are obtained in different limits: $q \to \infty$ and arbitrary finite $L$ for the SFF and $L \to \infty$ for arbitrary finite $q$ for the dynamics of correlation functions and entanglement.

## 4.3 T-dual impurity interactions

In the following we also compute the SFF and its moments in the semiclassical limit $q \to \infty$ for T-dual impurity interactions as well. That is, we choose two-qudit gates $U$ which remain unitary under computing its partial transpose with respect to either the first or the second qudit. An ensemble of such gates can be parameterized by [39, 56]

$$U = V(J)(u_0 \otimes u_1) \,, \tag{15}$$

with $V(J)$ a random diagonal two-qudit gate as well as local single qudit gates $u_0$ and $u_1$. More precisely, $V(J)$ mediates the interaction and has matrix elements

$$V(J)^{kl}_{ij} = \exp(\mathrm{i} J \xi_{ij}) \delta_{ik} \delta_{jl} \,, \tag{16}$$

in the product basis. The real parameter $J$ governs the interaction strength and the $\xi_{ij}$ are i.i.d. random variables with zero mean. The local unitaries $u_0$ and $u_1$ are taken to be independent Haar random unitaries from $U(q)$. The SFF is then defined by the average over both the local unitaries and the random phases $\xi_{ij}$. This choice of gates correspond to the random matrix ensembles introduced in Ref. [79] and with the local gates in the random phase circuit

introduced in Ref. [18]. We adapt the computation of the SFF and its moments in theses models outlined in Refs. [18,20,21] to the boundary chaos setting. The details of the computation can be found in App. A.2 and we only state the results here. The Haar average over $u_0$ and $u_1$ can again be expressed in terms of Weingarten functions as in the case of generic impurity interactions. While at $J = 0$ the model reduces to the toy model of Sec. 4.1, for non-zero interaction $J$ the average over the phases $\xi$ leads to non-trivial modifications of the factorized result in the toy model. To capture the influence of the interaction, let us denote by

$$\chi(J) = \left\langle e^{iJ\xi} \right\rangle_\xi,\tag{17}$$

the characteristic function of the random phases, where $\langle\cdot\rangle_\xi$ is the average over the distribution of the phases. Following Ref. [21], and keeping the interpretation of the boundary contributing the $m$-th moment of the SFF for a single particle system and the bulk contributing the $mn$-th moment at reduced time $t/n$ this allows for expressing the SFF and its moments as

$$K_m(t) = m! t^m \sum_{k=0}^{mn} A_k\left(\frac{t}{n}\right) |\chi(J)|^{2mt\left(1 - \frac{k}{mn}\right)},\tag{18}$$

where again $n = \gcd(L, t)$. The $A_k(x)$ are polynomials in $x$ of degree $mn$ and are given by [21]

$$A_k(x) = \sum_{l=k}^{mn} \binom{mn}{l}\binom{l}{k} [!(mn-l)] x^{mn-l}(x-1)^{l-k}.\tag{19}$$

Here $[!y]$ is the subfactorial. The polynomials obey the normalization condition $\sum_k A_k(x) = [(mn)!] x^{mn}$ such that at the non-interacting point $J = 0$, and hence $\chi(J) = 1$, Eq. (18) reproduces the SFF (13) of the non-interacting toy model as well as its moments. In fact the first term $m! t^m$ in Eq. (18) can be interpreted as the boundary's contribution to the full $m$-th moment of the SFF. On the other hand, the second term, i.e., the sum $\sum_k(\ldots)$, represents the contribution from the bulk, which in the presence of non-zero interaction $J$ gets suppressed compared to the toy model. More precisely, for nonzero $J$ and typical choices for the distribution of phases, one has $|\chi(J)| < 1$ causing all the terms in Eq. (18) except the term for $k = nm$ to be exponentially suppressed at late times. As moreover $A_{mn} = 1$ one has $K_m(t) \approx m! t^m$ for sufficiently large times implying that the SFF and its moments approach the random matrix result for the CUE after some non-universal initial dynamics. The latter time, after which the random matrix SFF is approached, defines the so-called Thouless time $t_{\text{Th}}$, i.e., the time scale of the onset of universal random-matrix like dynamics of the system. Equivalently, spectral correlations on quasi-energy scales below $\sim 2\pi/t_{\text{Th}}$ coincide with random-matrix spectral correlations. The (many-body) Thouless time is conventionally obtained from the SFF, $m = 1$ in Eq. (18). It can be roughly estimated by noting that $n \leq L$ and hence $|\chi(J)|^{2t(1-k/n)} \leq |\chi(J)|^{2t/L}$ as well as $\sum_k A_k(x) < \sqrt{2\pi L} t^L e^{1-L}$ using Stirling's approximation. This yields

$$K(t) \leq t + \sqrt{2\pi L} e^{1-L} t^{L+1} |\chi(J)|^{2t/L},\tag{20}$$

which indicates that non-universal corrections to the random matrix result initially grow (at most) like $t^{L+1}$. In contrast, at later times the non-universal corrections decay (at least) as $|\chi(J)|^{2t/L}$ and thus are expected to approach the random matrix result on extensive time scales. More precisely, Thouless time should scale (at most) as $t_{\text{Th}} \sim L^\mu$ for some positive $\mu$, not necessarily equal to one due to the $L$ dependence of the prefactor in the above bound. Formally we define the Thouless time as the time after which $|K(t) - t| \lesssim 1$. Solving this condition in terms of the $-1$ branch of the Lambert W-function and its asymptotic behaviour around zero yields

$$t_{\text{Th}} \lesssim \frac{L^2 \ln(L)}{|\ln|\chi(J)||},\tag{21}$$

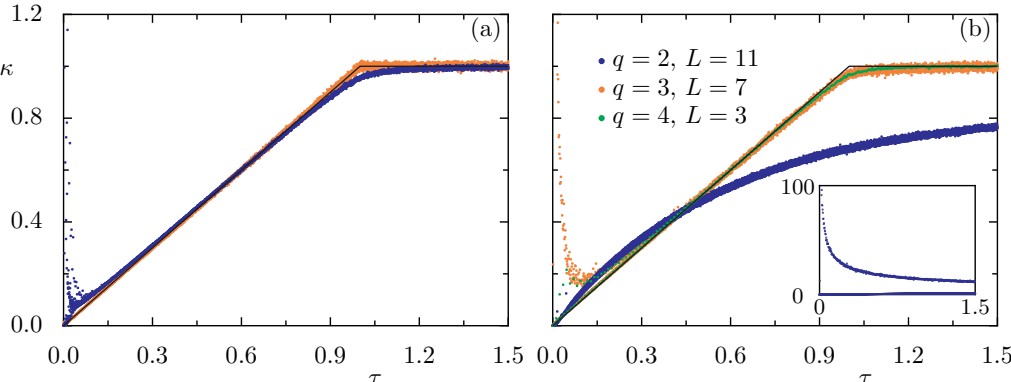

Figure 1: SFF for (a) Haar-random and (b) T-dual impurity interactions at $J = 3.1$ and uniformly distributed phases in $[-1, 1]$ for local Hilbert space dimensions (a) $q = 2, 3$ and system size $L = 11, 7$, as well as (b) $q = 2, 3, 4$ and $L = 11, 7, 3$ averaged over $> 10^4$ realizations. The black line corresponds to the random matrix result for the CUE. In the T-dual case at $q = 2$ the main panel depicts the SFF only for times, which are not integer multiples of $L = 11$. For the latter the SFF is enhanced by up to two orders of magnitude as shown in the inset, where we depict the SFF for all times.

for large $L$. It implies an upper bound for the Thouless time which scales essentially quadratic with system size, $\mu = 2$. Having non-zero Thouless time, one might call the system with T-dual impurity interactions "less chaotic" in the sense of spectral statistics as in the case of generic impurity interactions. In contrast, T-dual impurity interactions yield (local operator) entanglement growth at maximum speed [69] and fast thermalization [56], minding again that those properties were derived in a different limit than considered here.

## 5 Numerical observations

In this section we present numerical results for the SFF and its moments for small local Hilbert space dimension. In Fig. 1 we show the SFF for both (a) generic and (b) T-dual impurity interactions for local Hilbert space dimension $q = 2$ and $q = 3$ for the ensembles introduced in Sec. 4. For better comparison between different $q$ and different system sizes, we rescale both the SFF $\kappa = K/q^{L+1}$ and time $\tau = t/q^{L+1}$ by the respective Heisenberg time. For $q = 3$ the SFF matches the random matrix result for the unitary symmetry class well, possibly after some non-trivial initial dynamics. The latter are clearly seen on the shown scale in the T-dual case, while hardly visible in the generic case. Those initial deviations are particularly pronounced at times which are integer multiples of system size. The timescale after which they die out sets the Thouless time $t_{\text{Th}}$, which we find to scale with system size with a power law $t_{\text{Th}} \sim L^\nu$ where $\nu \approx 1$ and $\nu \approx 0.7$ for Haar random and T-dual imputity interactions, respectively; see below for details.

For $q = 2$ generic impurity interactions reproduce the CUE result as well after some non-trivial initial dynamics. In contrast for T-dual impurity interactions with $q = 2$ the SFF clearly deviates from the CUE result $K(t) = t$ ($\kappa(\tau) = \tau$). Instead it is drastically enhanced by up to two orders of magnitude at times which are integer multiples of system size (see inset), whereas it follows the COE (with adapted Heisenberg time $t_{\text{H}} \to \frac{L-1}{L} t_{\text{H}}$ [56]) otherwise. This hints towards an (weakly broken) anti-unitary symmetry present for T-dual impurity interactions at $q = 2$. In this case, the ensemble averaged level spacing distribution $p(s)$ shows even

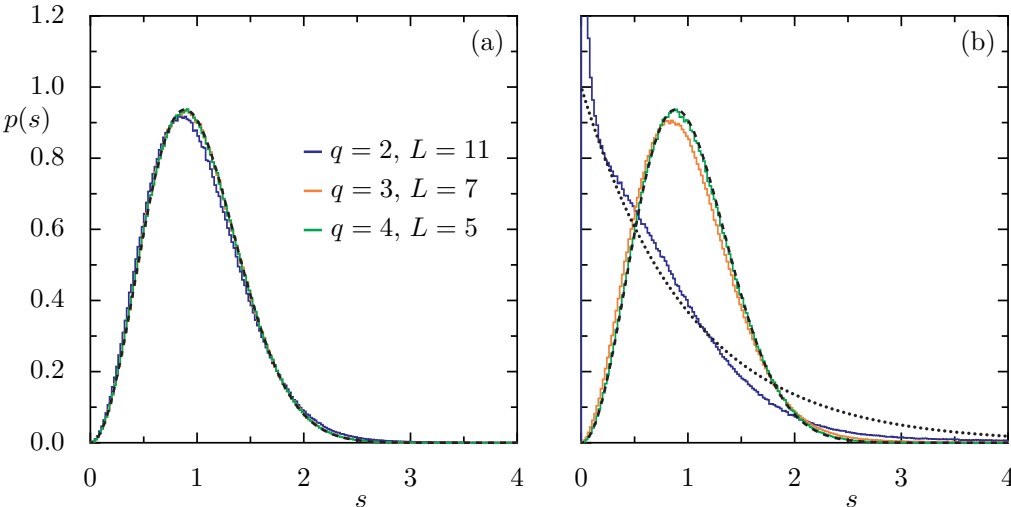

Figure 2: Ensemble averaged level spacing distribution for (a) Haar-random and (b) T-dual impurity interactions at $J = 3.1$ and uniformly distributed phases in $[-1, 1]$ for local Hilbert space dimensions $q = 2, 3, 4$ and system size $L = 11, 7, 5$ averaged over 500 realizations. The dashed and dotted black lines correspond to the Wigner-Dyson distribution for the CUE and to the exponential distribution of a Poissonian spectrum, respectively.

more dramatic deviations from the random matrix results for both the unitary and the orthogonal symmetry class, see Fig. 2(b). Instead $p(s)$ is closer to the exponential distribution of a Poissonian spectrum, but still differs significantly from the latter. In particular, we observe a large weight for small spacings with $p(s) \approx 4$ as $s \to 0$ (not captured in Fig. 2(b)). We attribute this highly degenerate spectrum to the fact, that for smallest $q = 2$ the probability for all phases $\xi_{ij}$ in Eq. (16) to be small is not negligible. Hence the model is with non-vanishing probability close to the non-interacting toy model even though the interaction parameter $J$ is large. This is in contrast with the transition oberved in the random phase circiut of Ref. [18], where Poissonian level spacings have been observed below a critical interaction parameter only. There, however, all the independent local gates are of the the form of the impurity interaction here and need to be close to the non-interacting point simultaneously. This becomes highly unlikely at larger interaction parameter, such that eventually the Wigner-Dyson level spacing distribution is restored. This is also expected in the present model for larger local Hilbert space dimension, for which the probability of being close to the non-interacting toy model becomes sufficiently small. Indeed, as depicted in Fig. 2, for local Hilbert space dimension $q \geq 3$ or generic impurity interactions at arbitrary $q$ the level spacing distribution is well described by Wigner-Dyson statistics for the CUE. Only for generic impurity interactions with $q = 2$ and T-dual impuritiy interactions with $q = 3$ small deviations are still visible, whereas for the respective next larger dimension the deviations disappear almost completely.

In the following we ignore the seemingly pathological case $q = 2$. We focus on local Hilbert space dimension $q \geq 3$, fixing $q = 3$, unless stated otherwise, instead and perform the required ensemble averages over $> 10^4$ realizations of the respective impurity interaction. In this situation the above numerical observations on the SFF exhibit similar phenomenology as predicted by the semiclassical results from Sec. 4. We discuss those numerical results in more detail and compare them with the above analytic description in the following sections.

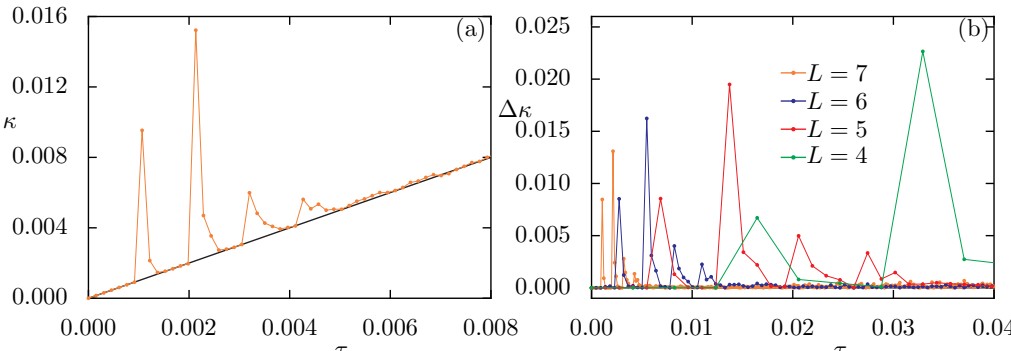

Figure 3: (a) SFF $\kappa$ for generic impurity interactions at initial times $\tau$ for $L = 7$ (connected symbols). The black line corresponds to the random matrix result. (b) Difference of SFF $\Delta\kappa$ from the random matrix result for $q = 3$ and various system sizes (connected symbols, see legend).

### 5.1 Haar-random impurity interactions

We start our discussion with the SFF and its moments for Haar-random impurity interactions. The analytical large $q$ result of Sec. 4.2, as well Fig. 1(a) indicate that the SFF follows the random matrix result for the CUE at all times. At small $q = 3$, however, we find deviations at initial times. This is illustrated in Fig. 3(a) which reveals deviations in the form of peaks at times $t$ which are integer multiples of system size. The semiclassical analysis suggests, that those peaks should vanish as $q \to \infty$ at fixed $L$. Additionally, our numerical analysis indicates, that upon proper rescaling by Heisenberg time both the height of the peaks and the time up to which they occur seem to vanish, when keeping $q$ fixed and increasing system size $L$. This is illustrated in Fig. 3(b), where we depict the difference of the numerically obtained SFF $\kappa(\tau)$ from the random matrix result, $\Delta\kappa(\tau) = \kappa(\tau) - \tau$ (for $\tau < 1$) for different system sizes. Without the rescaling, this implies that both the height of the peaks and the time scales up to which they occur grow slower than exponentially with the system size.

We comment on this in more detail below, but discuss higher moments of the SFF first. To this end we consider moments $\kappa_m(\tau)$ rescaled according to the exponential distribution predicted in the semiclassical limit and measured in units of Heisenberg time as $\kappa_m = q^{-(L+1)}(K_m/m!)^{1/m}$ as a function of $\tau$. Numerical results for the second and third moment are depicted in Fig. 4 and demonstrate excellent agreement with the random matrix result. Only for initial times large deviations in the form of sharp peaks occur around times, which are integer multiples of system size, similar as for the SFF. While their magnitude seems to grow with increasing system size at fixed $t/L$, those peaks decrease for later times at similar time scales as for the SFF. This sets Thouless time $t_{Th}$ and indicates the onset of universal, random-matrix like dynamics.

More precisely, we might define Thouless time $t_{Th}$ as the time after which $\Delta\kappa$ is smaller than a given small threshold. In Fig. 5(a) we depict $\Delta\kappa_m = \kappa_m - \tau$ (for $\tau < 1$) as a function of time $t/L$ for times, which are integer multiples of system size, i.e., for the times where we numerically observe the largest deviations of the SFF from random matrix theory. Due to the scaling of time by system size the curves almost coincide and, more importantly, all scale as approximately $\left(\frac{t}{L}\right)^{-\nu}$ with a fitted exponent $\nu \approx 4$. This implies Thouless time to scale linearly with system size, $t_{Th} \sim L$, and causes non-universal dynamics to cease after an extensive time scale. The second and third moment, shown in Fig. 5(a) and (c) respectively, show very similar features, including the same scaling behavior. Note, that from our semiclassical analysis we further expect Thouless time to decrease, when increasing $q$ while keeping $L$ fixed.

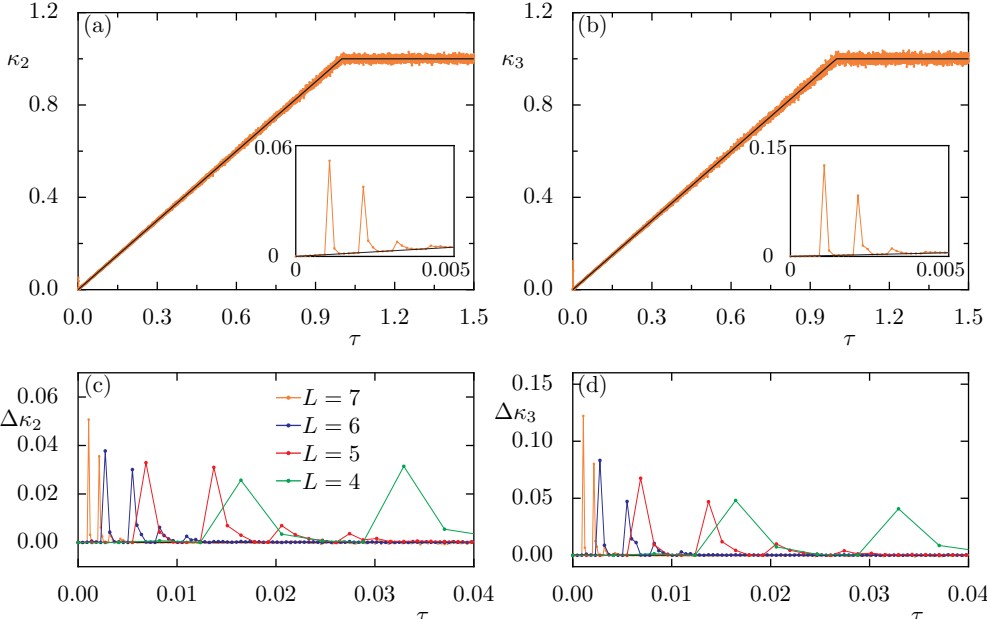

Figure 4: (a) Second($m = 2$) and (b) third ($m = 3$) moment of the SFF $\kappa_m$ for generic impurity interactions and $L = 7$ (connected symbols). The insets show magnifications at initial times. Black lines correspond to the random matrix result. (c) Difference $\Delta\kappa_2$ and (d) $\Delta\kappa_3$ from the random matrix result for initial times for various system sizes (connected symbols, see legend).

## 5.2 T-dual impurity interactions

We now focus on numerical results for T-dual impurity interactions. We choose phases $\xi_{ij}$ uniformly distributed in $[-1, 1]$ and fix the interaction parameter $J = 3.1$, ensuring chaotic dynamics. Figure 1(b) indicates that the SFF for $q = 3$ follows the random matrix result for sufficiently large times, whereas initial times show large deviations. The large $q$ results suggest that those deviations might be particularly pronounced, when $t$ and $L$ share a large common factor. This persists also for small $q$ as is shown in Fig. 6, where we depict the SFF for (a) times coprime with system size and find good agreement with the random matrix result for the largest accessible system sizes. In contrast, for (b) times, which are integer multiples of system size and hence $\gcd(L, t) = L$, we observe large deviations from the random matrix result at initial times. In case the greatest common divisor is smaller, also the deviations from the random matrix result decrease.

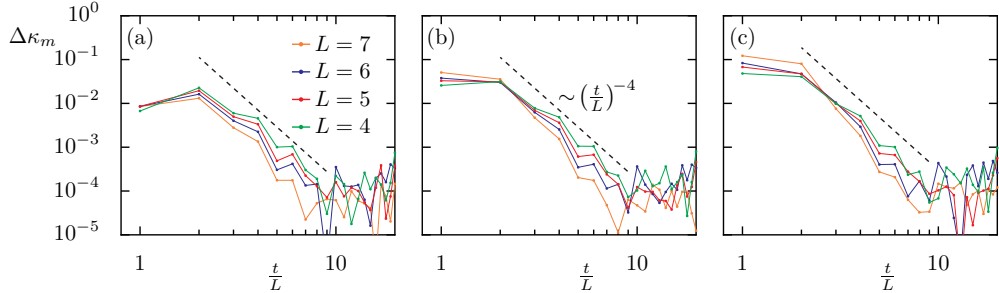

Figure 5: $\Delta\kappa_m$ vs $t/L$ for different system sizes and (a) $m = 1$, (b) $m = 2$ and (c) $m = 3$ (connected symbols, see legend). We only depict times, for which $t/L$ is an integer. The dashed black line indicates the scaling $\sim \left(\frac{t}{L}\right)^{-4}$ for $m = 1, 2, 3$.

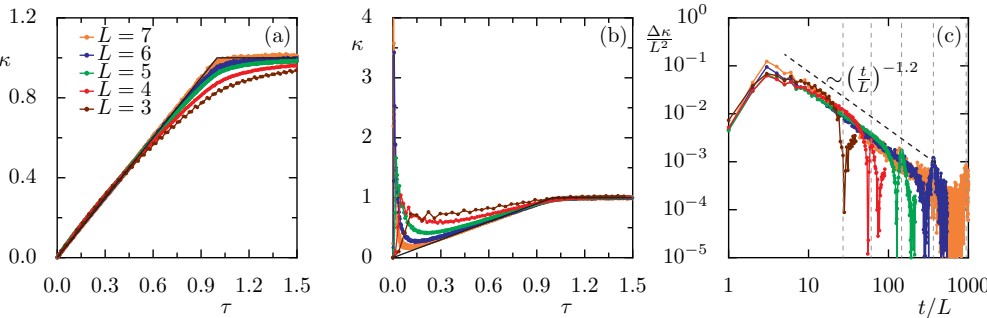

Figure 6: SFF $\kappa$ for various system sizes (connected symbols, see legend) for times (a) coprime with system size and (b) integer multiples of $L$. Black lines correspond to the random matrix result. (c) $\Delta\kappa/L^2$ vs. $t/L$. We only depict times for which $t/L$ is an integer, the gray dashed lines represent the respective Heisenberg times (corresponding to increasing $L$ from left to right). The dashed black line indicates the scaling $\sim \left(\frac{t}{L}\right)^{-1.2}$.

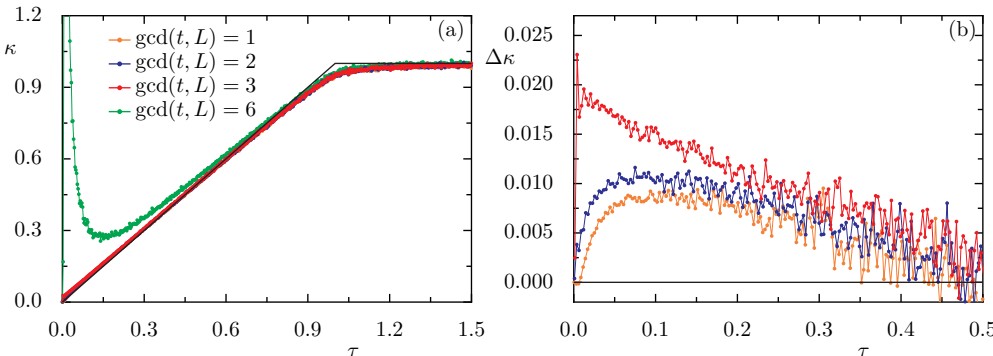

Figure 7: (a) SFF $\kappa$ vs $\tau$ for $L = 6$ grouped by the greatest common divisor $\gcd(t, L) = 1, 2, 3, 6$ (connected symbols, see legend). The black line indicates the random matrix result. (b) $\Delta\kappa$ vs. $\tau$ grouped by $\gcd(t, L) = 1, 2, 3$. Times, given as integer multiples of $L$ are omitted.

This motivates to use only those times with $\gcd(L, t) = L$ to estimate the Thouless time. In Fig. 6(c) we show the difference $\Delta\kappa$ from the random matrix result for these times. After scaling, the curves for $\Delta\kappa/L^2$ vs. $t/L$, approximately collapse (up to the respective Heisenberg time) and the scaled difference roughly decays as $(t/L)^{-\nu}$ with $\nu \approx 1.2$, with both the scaling of $\Delta\kappa$ by $L^2$ and the exponent obtained as a fit to the numerical data. Thouless time consequently scales as $t_{\text{Th}} \sim L^{\mu}$ with $\mu = (2 - \nu)/\nu \approx 0.7$. In general, the exponent $\mu$ might still depend on the parameter $J$ and on the local Hilbert space dimension $q$, as long as the latter is finite.

Moreover, in Fig. 7 we investigate the dependence on the SFF at time $t$ on $n = \gcd(L, t)$ in more detail. For $L = 6$, $n$ can take the values $1, 2, 3$ and $6$. We depict the SFF grouped by those possible values of $n$ in Fig. 7(a) and find SFF to be drastically enhanced at initial times for $n = L$, as predicted by the large $q$ results and as used for the extraction of the Thouless time. For the other possible values of $n$ the SFF is qualitatively similar, but an enhancement with larger $n$ still can be observed. This is shown in Fig. 7(b), where the difference $\Delta\kappa$, grouped by $n = 1, 2, 3$, is depicted. For larger $n$ also the SFF is larger.

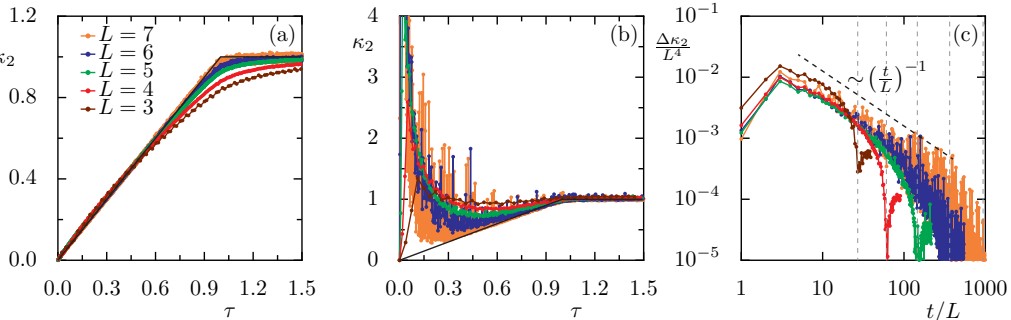

Figure 8: Second moment of the SFF $\kappa_2$ for various system sizes (connected symbols, see legend) for times (a) coprime with system size and (b) integer multiples of $L$. Black lines correspond to the random matrix result. (c) $\Delta\kappa_2/L^4$ vs. $t/L$. We only depict times for which $t/L$ is an integer, the gray dashed lines represent the respective Heisenberg times (corresponding to increasing $L$ from left to right). The dashed black line indicates the scaling $\sim \left(\frac{t}{L}\right)^{-1}$.

For the moments of the SFF we obtain very similar behavior. We illustrate this for the (rescaled) second moment $\kappa_2$, which we depict in Fig. 8. One of the main differences in comparison with the SFF are the larger fluctuations, particularly pronounced for larger $L$ and for $n = L$. This is a consequence of the number of realizations and we expect these fluctuations to decrease if more realizations of the impurity interaction are used. Additionally, the scaling of $\Delta\kappa_2(\tau) \sim L^4 (t/L)^{-1}$ differs. Hence, in contrast to generic impurity interactions, the second moment approaches the random matrix result at later times $\sim L^\mu$ with $\mu = (4 - \nu)/\nu \approx 3$, which is considerably later than $t_{\text{Th}}$ extracted from the SFF above. This can already be seen from comparing Fig. 6(b) with Fig. 8(b). For higher moments (not shown) we observe qualitatively similar results, with the approach to the random matrix result occurring at even later times due to the enhancement at times, which are integer multiples of system size.

We conclude this section by mentioning, that we expect the SFF for T-dual impurity interactions to be closer to the random matrix result for larger $q$. The latter, however, allows for numerical computations for even smaller system sizes only and is hence less suited for a numerical analysis. Nevertheless, we support our claim by depicting the SFF for $q = 4$ and $L = 3$ for the same interaction strength $J = 3.1$ as above in Fig. 9(a). The SFF shown there matches the random matrix result well, even for most times, which are integer multiples of $L$. Only for small time scales those times lead to visible deviations from the random matrix result as illustrated in Fig. 9(b). The enhancement occuring at those times is considerable smaller as for $q = 3$.

## 6 Conclusion

We studied spectral correlations in terms of the SFF in a minimal model of chaotic many-body quantum systems, previously dubbed boundary chaos, composed of a free, non-interacting quantum circuit, in which chaos and ergodicity is induced by an impurity interaction at the system's boundary. We obtained the SFF and all its moments exactly for two classes of impurity interactions (T-dual and generic) in the semiclassical limit of large local Hilbert space dimension. Analytical calculations become possible by "integrating out" the free bulk dynamics and thereby reducing the many-body to a two-body problem, which can be treated exactly at large $q$. The semiclassical results match with the expected random matrix result, after possibly non-universal initial dynamics in the case of T-dual impurity interactions. The latter is due to a

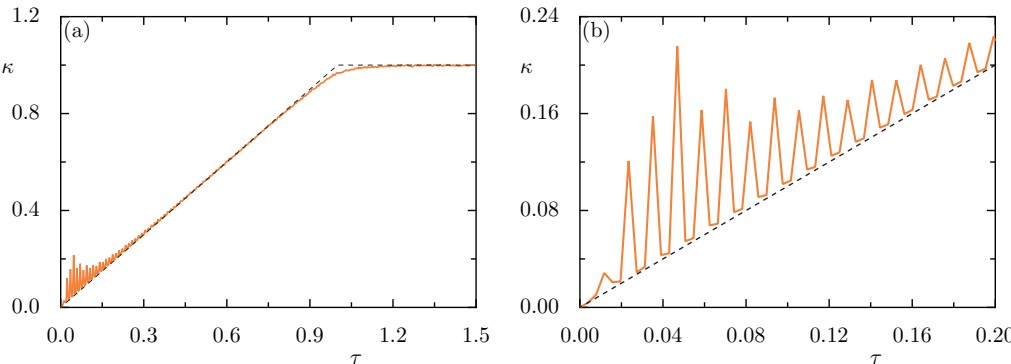

Figure 9: (a) SFF $\kappa$ vs. $\tau$ for $q = 4$ and $L = 3$ (orange line). Black lines correspond to the random matrix result. (b) is a magnification of (a).

resonance effect between time and system size, which causes the SFF to be enhanced when both integers share a large common factor. This effect is particularly pronounced for times, which are integer multiples of system size. Surprisingly, the large $q$ analytical results explain most of the phenomena observed at small $q$ in extensive numerical investigations. More precisely, the enhancement of the SFF at times which share a large common factor with system size is observed for small $q$ as well. In contrast to the semiclassical results generic impurity interactions at small $q$ show an enhancement of the SFF for times, which are integer multiples of system size, as well. That is, at small $q$ both classes of impurity interactions exhibit a non-zero Thouless time $t_{\text{Th}}$. The latter scales approximately as $L^{\mu}$ with system size, where $\mu \approx 0.7$ and $\mu \approx 1$ for T-dual and generic impurity interactions, respectively. In case of T-dual impurity interactions, higher moments approach the random matrix result at times later than $t_{\text{Th}}$.

Our work further consolidates the role of the boundary-chaos setting as a solvable minimal model for many-body quantum chaos, demonstrated here in terms of spectral statistics. In particular, our results imply, that in the Floquet setting a single impurity in an otherwise free system, i.e., in the absence of interactions in the unperturbed system, is sufficient to induce chaos. Interestingly, in a very loose sense, generic impurity interactions lead to spectral correlations closer to random matrix theory than T-dual impurity interactions, making the generic case "more chaotic". In contrast, T-dual impurity interactions lead to faster decay of correlations [56] and hence faster thermalization as well as faster growth of (local operator) entanglement [69] than generic impurity interactions, rendering the latter "less ergodic" in a similarly loose sense. Even though the above is a rough qualitative statement, it once again emphasizes that different notions of chaos and ergodicity in quantum systems are in general not equivalent. Hence care needs to be taken when handling those concepts in the quantum context and further research is necessary to unravel their connection.

## Acknowledgments

**Funding information**    FF acknowledges support by the Deutsche Forschungsgemeinschaft (DFG) Project No. 453812159. TP acknowledges the support by Program P1-0402 and Grants N1-0219 and N1-0233 of Slovenian Research and Innovation agency (ARIS). This project was partly funded within the QuantERA II Programme that has received funding from the European Union's Horizon 2020 research and innovation programme under Grant Agreement No. 101017733.

# A    Exact spectral form factor in the semiclassical limit

In this appendix we provide some details on the computation of the SFF and its moments in the semiclassical limit $q \to \infty$. As indicated in the main text this is achieved by exploiting the general theory of integrating monomials in the matrix elements over the unitary group developed in Ref. [76, 77]. Such integrals can be expressed in terms of Weingarten functions [76–78]. Their large $q$ asymptotics determines the SFF in the semiclassical limit. This has been used to compute the SFF in Refs. [17, 18, 21] and we adapt those methods in the following.

## A.1    Generic impurity interactions

We first consider generic impurity interactions, i.e., Haar random two-qudit gates.

**Spectral form factor**    By linearity of the expectation value we might average the terms in Eq. (11) individually. To this end, first note that the Haar average over an individual term

$$k(\mathbf{i}, \mathbf{j}, \mathbf{k}, \mathbf{l}) = \left\langle \langle \mathbf{ij} | U^{\otimes t} | \eta_1(\mathbf{i}) \, \eta_L(\mathbf{j}) \rangle \, \langle \mathbf{kl} | (U^\star)^{\otimes t} | \eta_1(\mathbf{k}) \, \eta_L(\mathbf{l}) \rangle \right\rangle, \tag{A.1}$$

is non-zero only if there are permutations $\sigma, \tau \in S_t$ such that $|\mathbf{ij}\rangle = |\sigma(\mathbf{k})\sigma(\mathbf{l})\rangle$ and $|\eta_1(\mathbf{i})\eta_L(\mathbf{j})\rangle = |\tau\eta_1(\mathbf{k})\tau\eta_L(\mathbf{l})\rangle$ or equivalently $|\mathbf{ij}\rangle = |\eta_1^{-1}\tau\eta_1(\mathbf{k})\eta_L^{-1}\tau\eta_L(\mathbf{l})\rangle$ [76,77]. For states $|\mathbf{i}\rangle$ and $|\mathbf{j}\rangle$ with pairwise distinct tensor factors the latter implies $\sigma = \eta_1^{-1}\tau\eta_1 = \eta_L^{-1}\tau\eta_L$. In the following we consider only such states, as those are almost all of the $q^{2t}$ basis states and all other states give only subleading contributions as $q \to \infty$. Note that in Eq. (A.1) the permutations $\sigma$ and $\tau$ act on both $\mathcal{H}_1^{\otimes t}$ and $\mathcal{H}_2^{\otimes t}$ simultaneously as the distinction between $\mathbf{i}$ and $\mathbf{j}$ is just an artifact of our choice for labeling the elements of the product basis in $\mathcal{H}_0 \otimes \mathcal{H}_1$. Any pair of such permutations contributes with $\mathrm{Wg}(\tau\sigma^{-1})$ to the average, such that [76,77]

$$k(\mathbf{i}, \mathbf{j}, \mathbf{k}, \mathbf{l}) = \sum_{\sigma, \tau \in S_t} \delta_{\mathbf{i}, \sigma(\mathbf{k})} \delta_{\mathbf{j}, \sigma(\mathbf{l})} \delta_{\mathbf{i}, \eta_1^{-1}\tau\eta_1(\mathbf{k})} \delta_{\mathbf{j}, \eta_L^{-1}\tau\eta_L(\mathbf{l})} \mathrm{Wg}\left(\tau\sigma^{-1}\right), \tag{A.2}$$

with the Kronecker $\delta$ being understood elementwise. For large $q$ the leading contribution to the above sum stems from $\mathrm{Wg}(\mathrm{id}) \sim q^{-2t}$, i.e., from $\sigma = \tau$. Consequently the leading contribution to the average comes from the translational invariant (invariant under conjugation by $\eta_1$) permutations. These are exactly the periodic shifts $\eta_r$ for $r \in \{0, 1, \ldots, t-1\}$. As any permutation that is invariant under conjugation by $\eta_1$ is automatically invariant under conjugation by $\eta_L$ the bulk of the system and its free dynamics do not yield any further constraints on the permutations contributing to the large $q$ asymptotics of the SFF. Keeping only the leading contributions as $q \to \infty$ given by $k(\mathbf{i}, \mathbf{j}, \mathbf{k}, \mathbf{l}) = q^{-2t} \sum_r \delta_{\mathbf{i}, \eta_r(\mathbf{k})} \delta_{\mathbf{j}, \eta_r(\mathbf{l})}$ we arrive at

$$K(t) = \sum_{\mathbf{ijkl}} k(\mathbf{i}, \mathbf{j}, \mathbf{k}, \mathbf{l}) = q^{-2t} \sum_{\mathbf{ijkl}} \sum_{r=0}^{t-1} \delta_{\mathbf{i}, \eta_r(\mathbf{k})} \delta_{\mathbf{j}, \eta_r(\mathbf{l})} = q^{-2t} \sum_{\mathbf{ij}} t = t. \tag{A.3}$$

Here, the sums runs only over the $\sim q^{2t}$ states $\mathbf{i}$ and $\mathbf{j}$ with pairwise distinct factors, which cancels the factor $q^{-2t}$. Hence, we recover the random matrix result for the CUE, i.e., the linear growth $K(t) = t$ in the limit $q \to \infty$. Note, that in this limit the Heisenberg time $q^{L+1}$ diverges as well, such that the plateau of the SFF for larger times cannot be resolved within this approach.

**Higher moments**    We now turn our attention to higher moments of the SFF, which are obtained by evaluating the Haar average of integer powers of Eq. (11). The structure of the arguments is very similar as above and we only sketch the main steps and refer to Refs. [20]

and [21] for details. To compute the $m$-th moment this involves averaging over $m$ replicas and hence a $4m$-fold sum over basis states $\mathbf{I} = (\mathbf{i}_n)_{n=1}^m$. We might think of the states $\mathbf{I} = (\mathbf{i}_n)_n = (i_{n,s})_{n,s}$ as a basis in $\mathcal{H}_1^{\otimes mt}$ and similar for $\mathbf{J} = (\mathbf{j}_n)_n$, $\mathbf{K} = (\mathbf{k}_n)_n$, and $\mathbf{L} = (\mathbf{l}_n)_n$. Permutations $\sigma \in S_{tm}$ again act on this bases by permuting tensor factors and we write, e.g., $|\sigma(\mathbf{I})\rangle$ for the basis state with accordingly permuted factors. To compute the $m$-th moment of the SFF, expressed as

$$K(t) = \sum_{\mathbf{I},\mathbf{J},\mathbf{K},\mathbf{L}} k(\mathbf{I},\mathbf{J},\mathbf{K},\mathbf{L}), \tag{A.4}$$

by rewriting Eq. (2), we average each term

$$k(\mathbf{I},\mathbf{J},\mathbf{K},\mathbf{L}) = \left\langle \prod_{n=1}^m \langle \mathbf{i}_n \mathbf{j}_n | U^{\otimes t} | \eta_1(\mathbf{i}_n) \, \eta_L(\mathbf{j}_n) \rangle \, \langle \mathbf{k}_n \mathbf{l}_n | (U^\star)^{\otimes t} | \eta_1(\mathbf{k}_n) \, \eta_L(\mathbf{l}_n) \rangle \right\rangle, \tag{A.5}$$

in the $4mt$-fold sum individually. Each of those terms is non-zero only if there are permutations $\sigma, \tau \in S_{tm}$ such that $|\mathbf{IJ}\rangle = |\sigma(\mathbf{K})\sigma(\mathbf{L})\rangle$ and $|\mathbf{IJ}\rangle = |(\eta_1^{\otimes m})^{-1} \tau \eta_1^{\otimes m}(\mathbf{K}) (\eta_L^{\otimes m})^{-1} \tau \eta_L^{\otimes m}(\mathbf{L})\rangle$ [76, 77]. Here $\eta_x^{\otimes m} \in S_{tm}$ denotes the permutation that implements the $t$-periodic shift by $x$ within each of the $m$ replicas. Expressing the Haar average in terms of Weingarten functions and using their large $q$ asymptotics again allows for evaluating Eq. (A.5) as $q \to \infty$. The permutations which contribute in this limit are those which are invariant under conjugation by both $\eta_1^{\otimes m}$ and $\eta_L^{\otimes m}$ with the latter not giving any further constraints. These permutations implement independent $t$-periodic shifts within each replica and additionally permute the replicas [20, 21]. They form a subgroup $G_m^{(t)}$ of $S_{tm}$ of order $|G_m^{(t)}(t)| = m! t^m$ and which for $m = 1$ reduces to the cyclic subgroup generated by $\eta_1$. In the semiclassical limit $q \to \infty$ we then obtain

$$k(\mathbf{I},\mathbf{J},\mathbf{K},\mathbf{L}) = q^{-2mt} \sum_{\sigma \in G_m} \delta_{\mathbf{I},\sigma(\mathbf{K})} \delta_{\mathbf{J},\sigma(\mathbf{L})}, \tag{A.6}$$

with the prefactor representing the asymptotic behavior of $\mathrm{Wg}(\mathrm{id}) \sim q^{-2mt}$. The $m$-th moment of the SFF consequently reads

$$K_m(t) = \sum_{\mathbf{I},\mathbf{J},\mathbf{K},\mathbf{L}} k(\mathbf{I},\mathbf{J},\mathbf{K},\mathbf{L}) = q^{-2mt} \sum_{\mathbf{I},\mathbf{J},\mathbf{K},\mathbf{L}} \sum_{\sigma \in G_m^{(t)}} \delta_{\mathbf{I},\sigma(\mathbf{K})} \delta_{\mathbf{J},\sigma(\mathbf{L})} = |G_m^{(t)}| = m! t^m, \tag{A.7}$$

where again the prefactor is canceled by the sum over the $\sim q^{2mt}$ states $\mathbf{I}$ and $\mathbf{J}$. The above coincides with the $m$-th moment for the CUE and the exponential distribution of the CUE SFF. That is, as $q \to \infty$ the boundary chaos circuit with Haar random impurity interactions reproduces the random matrix SFF and all its moments exactly.

## A.2   T-dual impurity interactions

In this section, we compute the SFF and its moments for T-dual impurity interactions drawn from the ensemble described in Sec. 4.3.

**Spectral form factor**   Again we begin by computing the average $k(\mathbf{i},\mathbf{j},\mathbf{k},\mathbf{l})$ defined in Eq. (A.1) for fixed $\mathbf{i}$, $\mathbf{j}$, $\mathbf{k}$, and $\mathbf{l}$, for which we again assume pairwise distinct tensor factors similar as in the previous section. For impurity interactions of the form (16) the average over the two local unitaries and the phases factorizes as

$$k(\mathbf{i},\mathbf{j},\mathbf{k},\mathbf{l}) = \left\langle \langle \mathbf{i} | u_o^{\otimes t} | \eta_1(\mathbf{i}) \rangle \, \langle \mathbf{k} | (u_o^\star)^{\otimes t} | \eta_1(\mathbf{k}) \rangle \right\rangle_{u_0} \left\langle \langle \mathbf{j} | u_1^{\otimes t} | \eta_L(\mathbf{j}) \rangle \, \langle \mathbf{l} | (u_1^\star)^{\otimes t} | \eta_L(\mathbf{l}) \rangle \right\rangle_{u_1}$$

$$\times \left\langle \exp\left( iJ \sum_{s=1}^t \left[ \xi_{i_s j_s} - \xi_{i_{s-1} j_{s-L}} \right] \right) \right\rangle_\xi. \tag{A.8}$$

Here the subscripts $s-1$ and $s-L$ are understood mod $t$ and the last bracket denotes the average over all the phases $\xi_{ij}$. We first consider the average over $u_0$, whose leading contribution coincides with the computation of the spectral form factor at time $t$ in a single particle system with Hilbert space dimension $q$. The average over the first qudit hence reads [18, 76, 77]

$$\left\langle \langle \mathbf{i}| u_o^{\otimes t} |\eta_1(\mathbf{i})\rangle \langle \mathbf{k}| \left(u_o^\star\right)^{\otimes t} |\eta_1(\mathbf{k})\rangle \right\rangle_{u_0} = \sum_{\sigma,\tau\in S_t} \delta_{\mathbf{i},\sigma(\mathbf{k})}\delta_{\mathbf{i},\eta_1^{-1}\tau\eta_1(\mathbf{k})}\mathrm{Wg}(\tau\sigma^{-1}) = q^{-t}\sum_{r=0}^{t-1}\delta_{\mathbf{i},\eta_r(\mathbf{k})}, \qquad \text{(A.9)}$$

where in the last equation we only keep the leading contributions as $q \to \infty$ similar to App. A.1.

Even though the first equality in the above equation holds also for $u_1$ with $\eta_1$ replaced by $\eta_L$, the average over $u_1$ is slightly more involved. This is due to the free bulk dynamics which couples local unitaries that are $L$ time steps apart, which might prevent unitaries at different time steps to be connected via intermediate steps. To be more precise, we denote by $n = \gcd(t, L)$ the greatest common divisor of time $t$ and systems size $L$ and introduce the integer $p = t/n$. This allows for writing

$$\langle \mathbf{j}| u_1^{\otimes t} |\eta_L(\mathbf{j})\rangle \langle \mathbf{l}| \left(u_1^\star\right)^{\otimes t} |\eta_L(\mathbf{l})\rangle = \prod_{k=1}^{n}\left(\prod_{s=1}^{p} \langle j_{k+sL}| u_1 |j_{k+(s-1)L}\rangle \langle l_{k+sL}| u_1^\star |l_{k+(s-1)L}\rangle\right), \quad \text{(A.10)}$$

with the subscripts again understood mod $t$. The average of this expression over $u_1$ can be interpreted as the average over $n$ disconnected replicas indexed by $k$, where each replica is given by the expression in parentheses. This is a consequence of $\eta_L$ being a product of $n$ disjoint cycles of length $p = t/n$. For a single replica the latter is just the expression to be averaged for the SFF of a single particle system at time $p = t/n$. Consequently, the average of Eq. (A.10) corresponds to computing the $n$-th moment of the SFF of a single $q$-dimensional CUE at time $p$, possibly after some irrelevant rearranging of tensor factors.[1] The leading contribution to the average [76, 77]

$$\left\langle \langle \mathbf{j}| u_1^{\otimes t} |\eta_L(\mathbf{j})\rangle \langle \mathbf{l}| \left(u_1^\star\right)^{\otimes t} |\eta_L(\mathbf{l})\rangle \right\rangle_{u_1} = \sum_{\sigma,\tau\in S_t} \delta_{\mathbf{j},\sigma(\mathbf{l})}\delta_{\mathbf{j},\eta_L^{-1}\tau\eta_L(\mathbf{l})}\mathrm{Wg}(\tau\sigma^{-1}), \qquad \text{(A.11)}$$

originates from permutations which are invariant under conjugation by $\eta_L$. Just as in App. A.1 these permutations implement independent periodic shifts (with period $p$) within each replica and additionally permute the replicas [20, 21]. They form a subgroup $G_n^{(p)}$ of $S_t = S_{pn}$ of order $|G_n^{(p)}| = n!p^n$. The average in the semiclassical limit $q \to \infty$ becomes

$$\left\langle \langle \mathbf{j}| u_1^{\otimes t} |\eta_L(\mathbf{j})\rangle \langle \mathbf{l}| \left(u_1^\star\right)^{\otimes t} |\eta_L(\mathbf{l})\rangle \right\rangle_{u_1} = q^{-t}\sum_{\sigma\in G_n^{(p)}}\delta_{\mathbf{j},\sigma(\mathbf{l})}. \qquad \text{(A.12)}$$

Note that in the case of co-prime $t$ and $L$ corresponding to $n = 1$ and $p = t$ the subgroup $G_1^{(p)}$ corresponds to the cyclic subgroup generated by $\eta_1$ and Eq. (A.12) reduces to Eq. (A.9).

It remains to compute the average over the phases $\xi_{ij}$. To this end we introduce

$$\Theta(\mathbf{i},\mathbf{j};\sigma,\tau) = \sum_{s=1}^{t}\left(\xi_{i_s j_s} - \xi_{i_{\sigma^{-1}(s)}i_{\tau^{-1}(s)}}\right), \qquad \text{(A.13)}$$

and note that [21]

$$\left\langle \exp\left(\mathrm{i}J\Theta(\mathbf{i},\mathbf{j};\sigma,\tau)\right)\right\rangle_\xi = |\chi(J)|^{2t-2\mathrm{fp}\left(\sigma^{-1}\tau\right)}, \qquad \text{(A.14)}$$

---

[1]In the following, such a rearrangement translates to going from $G_{mn}^{(p)}$ to one of its conjugacy classes. As the number of fixed points of a permutation does not change under conjugation this does not effect the final result for the SFF and its moments and we ignore this subtlety henceforth.

with $\chi(J) = \langle \exp(iJ\xi) \rangle_\xi$ the characteristic function of the distribution of the phases $\xi_{ij}$ and fp$(\sigma)$ the number of fixed points of the permutation $\sigma$. Putting this together we obtain for the SFF

$$K(t) = \sum_{\mathbf{ijkl}} k(\mathbf{i},\mathbf{j},\mathbf{k},\mathbf{l}) = q^{-2t} \sum_{\mathbf{ijkl}} \sum_{\sigma \in G_1^{(t)}} \sum_{\tau \in G_n^{(p)}} \delta_{\mathbf{i},\sigma(\mathbf{k})} \delta_{\mathbf{j},\tau(\mathbf{l})} |\chi(J)|^{2t-2\text{fp}(\sigma^{-1}\tau)} \tag{A.15}$$

$$= \sum_{\sigma \in G_1^{(t)}} \sum_{\tau \in G_n^{(p)}} |\chi(J)|^{2t-2\text{fp}(\sigma^{-1}\tau)}, \tag{A.16}$$

where in the last line the sum over the states cancels the asymptotic behavior of the Weingarten functions $\sim q^{-2t}$. By a change of summation variables the first sum trivializes and gives a factor $|G_1^{(t)}| = t$. The second sum can be simplified by noting that for any permutation in $G_n^{(p)}$ the number of fixed points is a multiple of $p = t/n$ and by introducing $A_k(p)$ as the number of permutations in $G_n^{(p)}$ with exactly $kp$ fixed points given by Eq. (19) [21]. This yields Eq. (18) for $m = 1$.

**Higher moments**    We now extend the above reasoning to the computation of higher moments. To this end we again use the basis $\mathbf{I} = (\mathbf{i}_k)_k = (i_{k,s})_{k,s}$ of $\mathcal{H}_1^{\otimes mt}$ with $k$ labeling the replicas and $s$ labeling the intermediate times. The generalization of Eq. (A.8) to the $m$-th moment reads

$$k(\mathbf{I},\mathbf{J},\mathbf{K},\mathbf{L}) = \left\langle \prod_{k=1}^{m} \langle \mathbf{i}_k | u_o^{\otimes t} | \eta_1(\mathbf{i}_k) \rangle \langle \mathbf{k}_k | (u_o^\star)^{\otimes t} | \eta_1(\mathbf{k}_k) \rangle \right\rangle_{u_0}$$

$$\times \left\langle \prod_{k=1}^{m} \langle \mathbf{j}_k | u_1^{\otimes t} | \eta_L(\mathbf{j}_k) \rangle \langle \mathbf{l}_k | (u_1^\star)^{\otimes t} | \eta_L(\mathbf{l}_k) \rangle \right\rangle_{u_1}$$

$$\times \left\langle \exp\left( iJ \sum_{k=1}^{m} \sum_{s=1}^{t} \left[ \xi_{i_{k,s} j_{k,s}} - \xi_{k,i_{s-1} j_{k,s-L}} \right] \right) \right\rangle_\xi. \tag{A.17}$$

In complete analogy to the computation of higher moments in App. A.1 the average over $u_0$ for large $q$ yields

$$\left\langle \prod_{k=1}^{m} \langle \mathbf{i}_k | u_o^{\otimes t} | \eta_1(\mathbf{i}_k) \rangle \langle \mathbf{k}_k | (u_o^\star)^{\otimes t} | \eta_1(\mathbf{k}_k) \rangle \right\rangle_{u_0} = q^{-mt} \sum_{\sigma \in G_m^{(t)}} \delta_{\mathbf{I},\sigma(\mathbf{K})}. \tag{A.18}$$

To compute the average over $u_1$ we observe that the $m$-fold product in the $u_1$ average in Eq. (A.17) in combination with the $n$-fold product in Eq. (A.10) allows for interpreting the $u_1$ average as the computation of the $mn$-th moment of the SFF at time $p = t/n$. The corresponding ensemble average is thus given by

$$\left\langle \prod_{k=1}^{m} \langle \mathbf{j}_k | u_1^{\otimes t} | \eta_L(\mathbf{j}_k) \rangle \langle \mathbf{l}_k | (u_1^\star)^{\otimes t} | \eta_L(\mathbf{l}_k) \rangle \right\rangle_{u_1} = q^{-mt} \sum_{\sigma \in G_{mn}^{(p)}} \delta_{\mathbf{J},\sigma(\mathbf{L})}. \tag{A.19}$$

It remains to perform the average over the phases $\xi_{ij}$. To this end let us denote by $\Theta(\mathbf{I},\mathbf{J};\sigma,\tau)$ the obvious generalization of Eq (A.13), which allows for expressing the $\xi_{ij}$ averages as

$$\left\langle \exp(iJ\Theta(\mathbf{I},\mathbf{J};\sigma,\tau)) \right\rangle = |\chi(J)|^{2mt-2m\text{fp}(\sigma^{-1}\tau)}. \tag{A.20}$$

Putting this together and repeating the computation which yields Eq. (18) we obtain.

$$K_m(t) = \sum_{\mathbf{IJKL}} k(\mathbf{I}, \mathbf{J}, \mathbf{K}, \mathbf{L}) = \sum_{\sigma \in G_m^{(t)}} \sum_{\tau \in G_{mn}^{(p)}} |\chi(J)|^{2t - 2\mathrm{fp}(\sigma^{-1}\tau)} \tag{A.21}$$

$$= m! \, t^m \sum_{k=0}^{mn} A_k \left(\frac{t}{n}\right) |\chi(J)|^{2mt\left(1 - \frac{k}{mn}\right)}, \tag{A.22}$$

with the $A_k$ computed for $G_{mn}^{(p)}$, as indicated in Eq. (19). The above is the result presented in Eq. (18) in the main text.

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
