# Peer review of "Boundary Chaos: Spectral Form Factor"

_SciPost Physics, doi:SciPost Phys. 17, 142 (2024)_

## Round 1 · Referee Report · Anonymous (Referee 1) · 2024-3-25

Strengths

  • Analytical results derived at large local Hilbert space dimension $q$ -Extensive numerical analysis

Weaknesses

-Main results are specific to boundary chaos

Report

In this work, the authors investigate the presence of quantum chaos in noninteracting circuits locally perturbed by an interacting gate (T-dual or Haar random) acting at the system’s boundary (boundary chaos). This is done through the analysis of the spectral form factor (SFF) and of its moments. In particular, results in the limit of large local Hilbert space dimension $q$ are obtained analytically by reducing the many-body problem to an effective two-body calculation. Their analytical findings at large $q$ agree with the expected results from random matrix theory (RMT) . The authors then compare these exact results to numerical simulations performed for small $q=2,3,4$. They show that the main features of SFF and its moments are captured by RMT also at small $q$ for times < Heisenberg time and > of a Thouless time, before which the SFF displays a non-universal behavior. Furthermore, their extensive numerical analysis gives an estimate of the Thouless time, depending on system size and the type of interacting gates perturbing the circuit.

The paper is well written, and the derived results are of interest for the community. I believe that the results contained in this manuscript deserve publication, and easily meet the acceptance criteria of SciPost Physics Core. However, my main concerns regarding the publication in SciPost Physics is that the in-depth analysis of SFF for boundary chaos does not provide general hints for other settings. I understand that this model serves as a minimal model for studying quantum many-body chaos, but I would appreciate if the authors can comment more on generic features that this study can highlight for other setting.

Some other minor comments are listed below.

-The authors may include the curve obtained with numerics for q=4 L=6 in Figure 1b and qualitatively comment on the observed deviations from RMT during the initial non-universal regime already at the beginning of Sec. 5.

-Similarly, it would be nice to add the results for q=4 L=6 in the plot of Figure 2b and show theconvergence towards a Wigner-Dyson distribution.

-For the Haar impurity, the authors extract a power-law dependence in t/L of the initial non-universal regime, with exponent $\nu\approx 4$ for the scaled SFF and for its moments. Is the value of this exponent obtained as a fitting parameter? Do the authors have an understanding for this value?

-Still on the data collapse for the T-dual impurity (Figure 6c). Is there an understanding of the factor $1/L^2$ in the scaled SFF needed to observe data collapse (and consequently of the Thouless time scaling exponent $\mu=(2-\nu)$)? Also, do the authors have an understanding —beyond the numerical evidence— on why the exponent $\mu$ for m=2 (Figure 8c) is different from that of the SFF? Do the authors have an estimate of the exponent $\mu$ for higher moments?

  • As the authors pointed out at the end of page 10, the Thouless time (for both types of impurities) it is expected to decrease when q is increased at fixed L. It would be nice to see how much the exponent $\nu$ changes for q=2,3 in the Haar case, and possibly $\mu$ for m=1,2 and q=3,4 in the T-dual.

---

## Round 1 · Referee Report · Anonymous (Referee 2) · 2024-6-28

Strengths

  1. Calculates the Spectral Form Factor analytically for a class of many-body models, also gives numerical results.
  2. Clearly written

Weaknesses

  1. Connections to physical models is not discussed sufficiently.

Report

The paper deals with a devilishly clever model of a many-body system that has elements of non-integrability while keeping the simplicity in the bulk via swap operators. Pushing all the complexity to the border, it is inspired by billiards: popular models of single particle chaos. The authors mention this at places and give insights into how a certain procedure is like the Poincare surface of section.

The primary quantity that is calculated is the spectral form factor (SFF) that forms an important tool to study chaos. The boundary gates that provide the single "impurity" are taken from (1) product gates, (2) T-dual gates: random unitaries times products and (3) Haar random. ary quantity that is calculated is the spectral form factor (SFF) that forms an important tool to study chaos.

As the bulk has swap operators, the authors are able to reduce the SFF calculation to a two-body problem each of local dimension q^t. For the product gate the SFF is found to be a higher power of t than for single CUE or two independent CUE. While for the random gate case the CUE result is reached for large q. The T-dual case is also like the random but is controlled by the coupling parameter and leads to a "thouless time".

The calculations seem fairly technical, but enough details are provided in the appendix for a determined reader.

Requested changes

  1. In Ref. 18, https://journals.aps.org/prl/pdf/10.1103/PhysRevLett.121.060601 the authors have found a phase transition (maybe MBL) as a function of what in the current paper is probably $J$. Maybe the authors can comment on it in the context of their model (esp. the T-dual case).

Recommendation

Publish (easily meets expectations and criteria for this Journal; among top 50%)

---

## Round 2 · Referee Report · Anonymous (Referee 3) · 2024-10-1

Report

The authors have made adequate changes in my opinion.

Recommendation

Publish (easily meets expectations and criteria for this Journal; among top 50%)

---

## Round 2 · Referee Report · Anonymous (Referee 4) · 2024-10-14

Report

I believe that the revised version of this manuscript incorporates the suggested changes and addresses the questions raised in my previous report, making parts of the manuscript clearer. I therefore recommend it for publication in SciPost Physics.

Recommendation

Publish (meets expectations and criteria for this Journal)

---

## Round 2 · Author Response

Reply to Report 1

We thank the Referee for their careful examination of our work and their overall positive eval-
uation. Thanks to their valuable suggestions, we improved the manuscript and addressed their
main concern regarding the unclear connection to more physical models (see details below). We
are convinced that the new version of our manuscripts deserves publication in SciPost Physics.

Referee: "However, my main concerns regarding the publication in SciPost Physics is that the in-
depth analysis of SFF for boundary chaos does not provide general hints for other settings. I
understand that this model serves as a minimal model for studying quantum many-body chaos,
but I would appreciate if the authors can comment more on generic features that this study can
highlight for other setting."

We added a comment on how our model relates to a body of existing work on integrabil-
ity breaking due to local perturbations in the Hamiltonian setting. There, both random matrix
spectral statistics and the onset of thermalization have been observed, similar as in the boundary
chaos setting. While the focus there is on perturbing interacting integrable systems, our study
reveals, that (at least in the Floquet case) interactions in the unperturbed model are not neces-
sary for the emergence of ergodicity and chaos. We admit, however, that the methods used for
obtaining our results do not carry over to interacting integrable models, i.e., even if the gates
in the bulk obey Yang-Baxter equation, the simplifications used in our approach are no longer
possible.

Referee: "The authors may include the curve obtained with numerics for q=4 L=6 in Figure 1b and
qualitatively comment on the observed deviations from RMT during the initial non-universal
regime already at the beginning of Sec. 5.
Similarly, it would be nice to add the results for q=4 L=6 in the plot of Figure 2b and show
the convergence towards a Wigner-Dyson distribution."

We added the data for q = 4 to Figure 1b and 2 and comment on the initial non-universal
regime to the beginning of Sec. 5.

Referee: "For the Haar impurity, the authors extract a power-law dependence in t/L of the initial non-
universal regime, with exponent ν = 4 for the scaled SFF and for its moments. Is the value of
this exponent obtained as a fitting parameter? Do the authors have an understanding for this
value? [...] Still on the data collapse for the T- dual impurity (Figure 6c). Is there an understanding
of the factor 1/L2 in the scaled SFF needed to observe data collapse (and consequently of the
Thouless time scaling exponent ν = (2 − ν))? Also, do the authors have an understanding -
beyond the numerical evidence - on why the exponent µ for m = 2 (Figure 8c) is different from
that of the SFF? Do the authors have an estimate of the exponent µ for higher moments?"

The observed scaling behavior and L dependence is obtained purely by a (rough) fit of the
data. We clarify this point now in the main text. We currently do not have an explanation for the
observed scaling behavior, neither from analytical arguments (finite q corrections to our
results are notoriously hard to obtain) nor from a simplified intuitive picture.

Referee: "As the authors pointed out at the end of page 10, the Thouless time (for both types of
impurities) it is expected to decrease when q is increased at fixed L. It would be nice to see how
much the exponent ν changes for q = 2, 3 in the Haar case, and possibly µ for m = 1, 2 and
q = 3, 4 in the T-dual."

This is in fact expected only in the case of Haar random impurities, as the Thouless time
there is finite ∼ L at q = 3 but vanishes in the limit q → ∞. In contrast, for T-dual impurities,
Thouless time scales as ∼ L^0.7 at q = 3 while it scales as ∼ L^2 as q → ∞, implying larger
Thouless time for sufficiently large L. However, as for fixed L only a few small values of q are
numerically accessible. We do not expect to be able to extract a clear scaling of the exponents µ
and ν.

Reply to Report 2

We thank the Referee for the evaluation of our work and the recommendation for publication in
SciPost Physics. We nevertheless addressed their main criticism by adding a paragraph on the
connection of our setup to physical models as well as with the model from Ref. [18], see below.

Referee: "Connections to physical models is not discussed sufficiently."

We added a paragraph to the introduction, relating our model to other work on locally per-
turbed (interacting) integrable systems, highlighting that in our case the bulk is non-interacting
and yet we observe the onset of chaos.

Referee: "In Ref. 18, https://journals.aps.org/prl/pdf/10.1103/PhysRevLett.121.060601 the authors
have found a phase transition (maybe MBL) as a function of what in the current paper is proba-
bly J. Maybe the authors can comment on it in the context of their model (esp. the T-dual case)."

Such a transition does not occur in our setting at q = 2 as we observe a level-spacing
distribution closer to Poissonian statistics even for large interaction strength J. We attribute
this to the model having non-negligible probability to be close to a non-interacting model. We
added a corresponding comment on this in the main text, when discussing the level spacing
distribution.

---

## Round 2 · List of Changes

- Page 2, first paragraph: Add comment on connection to physical models and relevant references
- Page 9, first paragraph: add comment on Thouless time
- End of Page 9 - beginning of page 10: Comment on (lack of) transition in the level spacing distribution p(s) and short description of p(s) for q=4
- Page 11, final paragraph of Sec. 5.1.: exponent -> fitted exponent
- Page 13: middle paragraph: add comment on the fitting of numerical data
- Page 15: final paragraph: highlight the difference of the model to the physical models introduced on page 2
- Figure 1(b) add curve for q=4
- Figure 2 add curve for q=4 in (a) and (b) respectively

---

## Editorial Decision

published